# Urgent Coronary Artery Bypass Grafting Complicated by Systemic Inflammatory Response from Fulminant Herpes Zoster Successfully Managed with Adjunct Extracorporeal Hemoadsorption: A Case Report [note 1]

**DOI:** 10.3390/jcm11113106

**Published:** 2022-05-31

**Authors:** Zaki Haidari, Wilko Weißenberger, Bartosz Tyczynski, Ender Demircioglu, Efthymios Deliargyris, Martin Christ, Matthias Thielmann, Mohamed El Gabry, Arjang Ruhparwar, Daniel Wendt

**Affiliations:** 1Department of Thoracic and Cardiovascular Surgery, West German Heart and Vascular Center, 45122 Essen, Germany; zaki.haidari@uk-essen.de (Z.H.); wilko.weissenberger@uk-essen.de (W.W.); ender.demircioglu@uk-essen.de (E.D.); matthias.thielmann@uk-essen.de (M.T.); mohamed.elgabry@uk-essen.de (M.E.G.); arjang.ruhparwar@uk-essen.de (A.R.); 2Department of Nephrology, University Hospital Essen, 45147 Essen, Germany; bartosz.tyczynski@uk-essen.de; 3Cytosorbents Inc., 305 College Road East, Princeton, NJ 08540, USA; edeliargyris@cytosorbents.com; 4Department of Cardiology and Intensive Care Medicine, Knappschaftskrankenhaus Bottrop, 46242 Bottrop, Germany; martin.christ@kk-bottrop.de

**Keywords:** hemoadsorption, viral disease, cardiac surgery, sepsis

## Abstract

Blood purification by hemoadsorption therapy seems to improve outcomes in selected patients undergoing cardiac surgery with cardiopulmonary bypass. Here, we report the successful application of hemoadsorption in the severe systemic inflammatory response during coronary artery bypass surgery in a patient with reactivated herpes zoster.

## 1. Introduction

The application of hemoadsorption therapy in cardiac surgery is increasing, as data have shown clinical benefits during complex procedures with a high risk for systemic inflammatory response or sepsis [1]. This is the first case in which hemoadsorption therapy has been used due to a severe systemic inflammatory response in combination with active herpes zoster during an urgent coronary artery bypass grafting with cardiopulmonary bypass (CPB).

## 2. Case Description

A 56-year-old man presented with unstable angina pectoris and stenosis of the left main. The patient’s history included arterial hypertension and severe chronic obstructive pulmonary and peripheral artery disease. Coronary angiography revealed severe three-vessel disease with high-grade stenoses of the left main, the proximal left, the anterior descending, the ramus intermedius coronary artery and an occluded right coronary artery. Transthoracic echocardiography showed a good left ventricular function without any sign of valvular dysfunction. Preoperative workup revealed proximal stenosis of the left subclavian artery with subclavian steal syndrome. During physical examination at admission, well-defined grouped vesicles could be identified on an erythematous background in a segmental nerve distribution of the left thorax (Th2-3, Figure 1). After dermatologic consultation, a clinical diagnosis of herpes zoster was made and antiviral therapy with intravenous acyclovir in combination with analgesics and topical therapy was started. The diagnosis was based on the classical clinical presentation and cutaneous findings. The patient was under continuous monitoring and intravenous heparin and nitroglycerin therapy in the ICU and was evaluated by the dermatologists every second day. After clinical and inflammatory improvement (eight days later), CABG surgery was performed. However, immediately after induction of anaesthesia, hemodynamic instability developed directly after induction of anaesthesia with noradrenergic support of >0.75 µg/kgbw/min (90 kg body weight, 2 mg/50 mL at 100 mL/min. flow rate) at a systolic blood pressure of 80 mmHg. Moreover, this hemodynamic instability further worsened after going on-pump with noradrenaline requirements reaching 1.5 µg/kgbw/min. At this juncture, the decision was made to integrate the hemoadsorption device CytoSorb^™^ (Cytosorbents, Monmouth Junction, NJ, USA) on the CPB circuit. Figure 2 exemplifies the CytoSorb hemoadsorption device and the possible ways it can be implemented in different extracorporeal circuits). Despite maximal supportive therapy (1000 mg cortisone IV, 2mg clemastin IV, volume therapy, and an additional dose of 500 mg acyclovir), high vasopressor support (around 1.0 µg/kgbw/min. throughout the CPB-time) was required during and after weaning from CPB. The fluid balance after weaning from CPB was 8500 mL on the positive side. In addition, the patient also received significant transfusions (8 units of RBC and 4 of platelets) and coagulation factor replenishments (5000 IE of prothrombin complex concentrate and 16g fibrinogen). An intraoperative echocardiography post-CPB revealed good left ventricular function and transit-time flow measurement of the grafts demonstrated excellent coronary flows. Veno-arterial extracorporeal membrane oxygenation (ECMO) therapy was necessary to reduce the vasopressor needs. The postoperative course was further complicated by disseminated intravascular coagulation requiring blood products and coagulation factors. Renal replacement therapy including continuing hemoadsorption was also introduced (three additional devices were used until postoperative day 4). After hemodynamic stabilization and decreasing lactate and inflammatory parameters, veno-arterial ECMO therapy was removed on the second postoperative day. The IL-6 and PCT values reached maximum levels of 66.745 pg/mL and 344 ng/mL, respectively. The changes in inflammatory mediators, lactate levels, and vasopressor needs over the three additional days of hemoadsorption are shown in Figure 3. The patient developed acute respiratory distress syndrome and required veno-venous ECMO therapy on postoperative day 3. Following a gradual respiratory improvement, veno-venous ECMO was explanted on the 15th postoperative day. A broncho-alveolar lavage did not identify any evidence of herpes simplex or COVID-19 infection. After percutaneous tracheostomy, weaning from mechanical ventilation was started. At 6 months follow-up, the patient is asymptomatic and active.

## 3. Discussion

Removal of cytokines by hemoadsorption in patients with sepsis or systemic inflammatory response syndrome has recently gained increased interest in cardiac surgery [1]. However, patient selection and timing of application have become very important in this adjunctive therapeutic strategy [2]. In the current case, a severe systemic inflammatory response with significant clinical deterioration occurred directly after induction of anaesthesia triggering the decision to initiate intraoperative hemoadsorption once the patient was placed on CPB. Postoperatively, we observed an unusually exaggerated increase of procalcitonin (PCT) and interleukin 6 (IL-6) and an ongoing need for inotropic support and decided to continue postoperative hemoadsorption therapy until day 4. During the course of the therapy, there was a gradual reduction in the circulating inflammatory markers that coincided with clinical improvement suggesting that hemoadsorption was an important contributor to the favorable outcome. 

The CytoSorb™ device has also been used as an adjunct therapy in critically ill COVID-19 patients with ARDS. Several published reports, including the multicenter U.S. CTC registry (NCT04391920) have reported favorable outcomes and high survival rates in high-risk COVID-19 patients being treated with simultaneous hemoadsoprtion during veno-venous ECMO [3]. In contrast, a small single-center study reported higher mortality in patients being treated with CytoSorb^™^ on VV-ECMO; however, it is difficult to draw any conclusions from this severely underpowered (17 patients per group) trial [4]. To our knowledge, this is the first case where hemoadsorption was used in a patient with herpes zoster who underwent urgent cardiac surgery complicated by severe intra- and postoperative hemodynamic deterioration and ARDS requiring prolonged ECMO support.

One limitation deserving mention is that only IL-6, PCT, and CRP were measured as markers of the severe inflammatory response in the present case, whereas numerous other markers may also be informative in this setting (e.g., IL-1β, IL-18, TNF-α, IL-10, and HMGB-1). Nevertheless, the presence of a cytokine storm is substantiated due to the fact that IL-6 levels in this case far exceeded the 15pg/mL upper reference limit of our laboratory.

More evidence is needed to better define the value of hemoadsorption in cardiac surgery especially relating to appropriate patient selection and timing and dosing of application [2,5]. In 2016, Bernardi et al. evaluated intraoperative hemoadsorption in elective CABG patients with no differences in regard to cytokine release [6]. This holds true, as in elective CABG patients, without any increased risk for inflammation, intraoperative hemoadsorption seems not to be justified. Just recently, the randomized controlled REMOVE study has been published evaluating intraoperative hemoadsorption in endocarditis patients showing no benefit on the delta pre- to postoperative SOFA score [7]. However, the REMOVE trial could show that all-important cytokines, including cell-free DNA, were highly significantly reduced in the hemoadsorption group (first 50 randomized patients being evaluated). Others have reported different results in regard to sepsis-associated mortality and postoperative SOFA scores, especially in the setting of endocarditis [1,8]. 

## 4. Conclusions

This case suggests that hemoadsorption may be a vital adjunct therapeutic option for the management of a profound systemic inflammatory response in a patient requiring urgent cardiac surgery. From a clinician standpoint, the pros and cons for hemoadsorption are the following: Pros: (1) easy to implement in various extracorporeal circuits, (2) proven safety, and (3) potential preventive effect in cardiac surgery if implemented directly into CPB. Cons: (1) more data needed (patient selection, timing and dosing), (2) additional costs, and (3) the potential removal of antibiotics or beneficial substances. Therefore, future evaluations are needed for a better and more profound understanding of hemoadsorption. Recently, a dedicated risk score (CytoScore) has been proposed to help select the right patients and determine the right time for hemoadsorption initiation that may prove very useful if validated in future studies [9]. Moreover, in other specialties outside the cardiovascular field, hemoadsorption also has the potential to expand away from CPB by putting patients pre-operatively on a hemoperfusion system with an integrated adsorber attached (traumatology, burn injury, neurosurgery, antithrombotic removal).

## Figures and Tables

**Figure 1 jcm-11-03106-f001:**
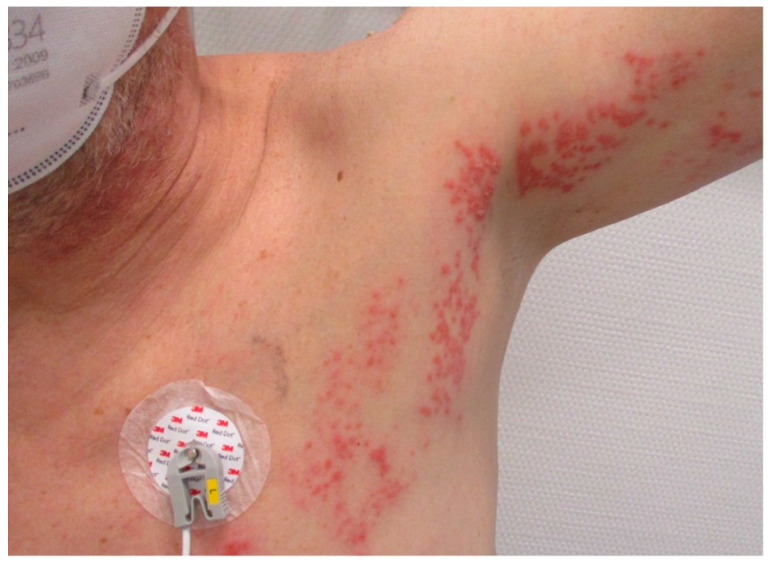
Herpes zoster infection, pathognomonic clinical appearance.

**Figure 2 jcm-11-03106-f002:**
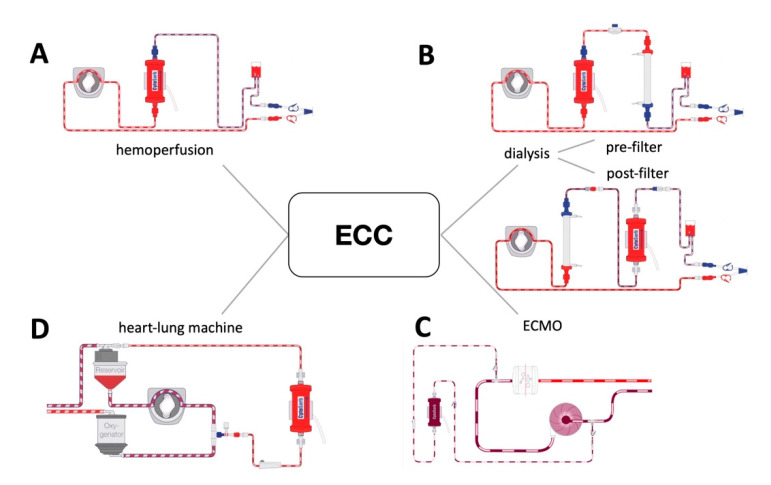
Different possibilities to implement Cytosorb in various extracorporeal circuits. (**A**): integration of CytoSorb into a hemoperfusion circuit; (**B**): integration of CytoSorb in dialysis circuit, pre- or post-filter; (**C**): integration of CytoSorb into heart-lung-machine; (**D**): integration of CytoSorb into an ECMO circuit.

**Figure 3 jcm-11-03106-f003:**
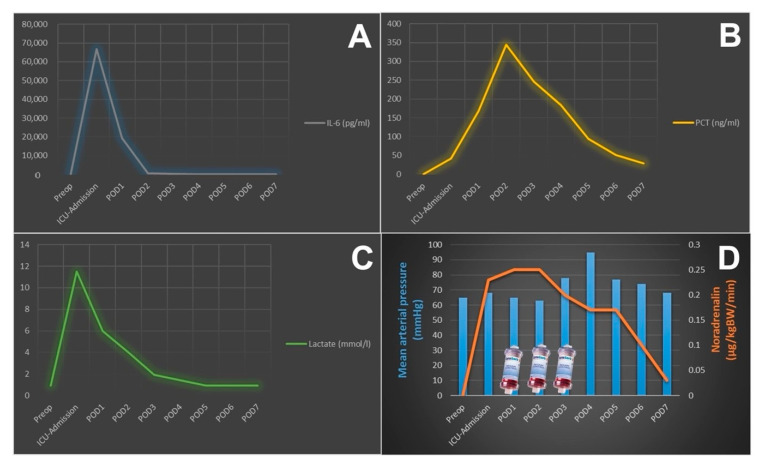
Postoperative course of inflammatory parameters, lactate, and vasopressor needs: (**A**) course of IL-6 (pg/mL), (**B**) Course of PCT (ng/mL), (**C**) Course of lactate (mmol/L), (**D**) Course of norepinephrine (µg/kgBW/min, orange line) and mean arterial blood pressure (blue bars) and three additional adsorber treatments on POD 1, 2 and 3.

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
