# Peer review of "Urgent Coronary Artery Bypass Grafting Complicated by Systemic Inflammatory Response from Fulminant Herpes Zoster Successfully Managed with Adjunct Extracorporeal Hemoadsorption: A Case Report [Author-notes fn1-jcm-11-03106]"

_jcm, 2022, doi:10.3390/jcm11113106_

Round 1

Reviewer 1 Report

- Overall, there is no presentation of specific values in terms of dosage, hemodynamics, etc. as below. This makes it difficult to cite the treatment in this paper for similar cases.

line 42

How unstable is the "instability" indicated by "Hemodynamic instability occurs directly after induction of anesthesia"?

line 45

"Despite maximal supportive therapy" is specifically how much was used?

line 46

What exactly is "high vasopressor support?"

- The reviewer would like authors to present the hemodynamic changes during the use of CytoSorb on a time course in Figure.

- If possible, a picture of CytoSorb would be helpful in imagining the treatment.

- With the biomarkers authors present, can authors say that CytoSorb was effective in this disease?

Author Response

Comments and Suggestions for Authors

Overall, there is no presentation of specific values in terms of dosage, hemodynamics, etc. as below. This makes it difficult to cite the treatment in this paper for similar cases.

We are thankful for this remark. We have now added more information on hemodynamics and inotropic support into the revised version of the case report.

line 42

How unstable is the "instability" indicated by "Hemodynamic instability occurs directly after induction of anesthesia"?

We are grateful for this remark. We went back into the anesthetic protocol and revised the case report accordingly. We calculated the values in µg/kgbw/min. to be more precise according to body weight and to be better comparable.

"Despite maximal supportive therapy" is specifically how much was used?

We appreciate this remark and now, in the revised version of the case report, we have now added more detailed information about supportive therapy. This has now been indicated and better explained with more detail.

What exactly is "high vasopressor support?"

We completely agree with your comment being to un-precise. This has now been changed in the revised version of the case report. In this particular case, cumulative inotropic support was above 1.0 µg/kgbw/min.

The reviewer would like authors to present the hemodynamic changes during the use of CytoSorb on a time course in Figure.

We greatly appreciate this comment and we added a new figure with the hemodynamics changes (mean arterial pressure) into the revised version of the case report. Moreover, we added a figure of CytoSorb for better understanding.

- If possible, a picture of CytoSorb would be helpful in imagining the treatment.

- With the biomarkers authors present, can authors say that CytoSorb was effective in this disease?

We greatly appreciate this well-founded remark. As of now, there is no standardized biomarker panel being present to control especially the septic component and situation of this very specific disease especially in cardiac surgery. Most authors proposed IL-6, TNFa or PCT as biomarkers. As to control for this serious situation, we focused mainly on IL-6 and PCT measurements. This has now been discussed in the revised version. Moreover, to rule out for others causes, we routinely performed BALs (broncho-alveolar lavage) on the ICU to further screen for HSV or COVID-19. This has now also implemented in the revised version.

Reviewer 2 Report

I have read thoroughly your case report on hemoadsorption i coronary by-pass surgery.  From the lingivistic an formal point the article is well written, but I am struggling to find the real novelty in your case and there are other even more important issues to be addressed.

  1. First the hemoadsorption as a method was studied in numerous case reports and trials in patients with systemic inflamatry disease, most recently in COVID-19 patients (viral disease with similar SIRS reaction as you described).
  2. Moreover, the hemoadsorption was also studied in the coronary-artery by-pass surgery scenario (Bernardi MH et al. Effect of hemoadsorption during cardiopulmonary bypass surgery – a blinded, randomized, controlled pilot study using a novel adsorbent) or more importantly in cardiac surgery for infectious endocarditis (Diab M et al. Cytokine hemoadsorption during cardiac surgery versus standard surgical care for infective endocarditis (REMOVE) and other.
  3. I would also expect that you describe more thoroughly the cytokine storm (not only by IL-6 but also by other usefull cytokines as IL-1β, IL-6, IL-18, TNF-α, and IL-10 and HMGB-1.
  4. The diagnosis of herpes zoster though probably correct is not supported by blood sampling for virus antibodies, why? The titer might be interesting
  5. Of, note the structure of the manuscript text is devided in sections but the content does not always correspond.
  6. I am completely missing a real discussion on pros and cons of hemoadsorption discussing the pathopysiology and clinical outcomes.
  7. Some sound conclusion with practical tips is also missing

In its present form I am not convinced that this case report paper will add some new information to the body of scientific literature. It may be reconsidered for publication after complete rewriting as a case report and review of literature.

Author Response

We have to thank the reviewer for this honest review, especially, that the CR is well-written from a linguistic point. In summary, we totally agree with all comments and to clarify the novelty your remarks, we will answer your questions thoroughly in the following and we have now implemented the discussion of the current literature as requested:

  1. We totally agree with this reviewer comment. Of course, the CR has been evaluated especially in COVID-19 patients, but has not been evaluated in herpes zoster so far. Nevertheless, we have now implemented a corresponding part in regard to COVID-19 into the revised version of the discussion section.
  2. We appreciated the reviewer comment and we have now implemented the two mentioned references and both have now been discussed in the revised version of the case report.
  3. We greatly appreciate this well-founded remark. We went back into lab results, but according to our institutional protocol, only IL-6 and PCT have been evaluated. Nevertheless, we totally agree, that especially such additional biomarkers might be of extreme interest and therefore, we modified our discussion part, mentioning this issue as a limitation and adding the above-mentioned biomarkers.
  4. We are thankful for this comment. Usually, herpes zoster represents only a look diagnosis, as it was in this case. The patient was evaluated every second day during the pre-operative course. We went again to all digital consultations in our local PDMS system in the department of dermatology and they did not any kind of specific lab testing, as the clinical signs and look diagnosis was pathognomonic. This has now been added in the revised version of the case report.
  5. The manuscript has been updated accordingly
  6. We are grateful for this well-founded remark, and now, the discussion part has been completely modified included all mentioned references above.
  7. We really appreciate this from a clinician well-founded remark and to give some conclusion and practical tips, we have modified the manuscript including a conclusion part with pros and cons of hemoadsorption.

Round 2

Reviewer 1 Report

This revision has improved many concerns.

However, the various perioperative treatment details are difficult to understand only by the text, and the additional information regarding the various perioperative treatments over time in the figure will help us to understand the effectiveness of CytoSorb better.

Author Response

We have to thank the reviewer for this excellent idea and therefore, we have added this additional information into Figure #3.

Reviewer 2 Report

Dear authors, you have provided many changes to the original manuscript. Most of them appropriate according to the content, spell check is highly recommended however (missing letters - nevetheless etc., inappropriate prepositions).

Author Response

We have to thank the reviewer for this remark, and we have now corrected the language and did an extensive spell check. The final version is now updated accordingly.